# Chronic Hepatitis B Infection: New Approaches towards Cure

**DOI:** 10.3390/biom13081208

**Published:** 2023-08-01

**Authors:** Mojisola Ogunnaike, Srijanee Das, Samiksha S. Raut, Ashrafi Sultana, Mohammad Ullah Nayan, Murali Ganesan, Benson J. Edagwa, Natalia A. Osna, Larisa Y. Poluektova

**Affiliations:** 1Department of Pharmacology and Experimental Neuroscience, University of Nebraska Medical Center, Omaha, NE 68198, USA; mogunnaike@unmc.edu (M.O.); srijanee.das@unmc.edu (S.D.); sraut@unmc.edu (S.S.R.); asultana@unmc.edu (A.S.); mohammadullah.nayan@unmc.edu (M.U.N.); murali.ganesan@unmc.edu (M.G.); 2Department of Pathology and Microbiology, University of Nebraska Medical Center, Omaha, NE 68198, USA; 3Department of Internal Medicine, University of Nebraska Medical Center, Omaha, NE 68198, USA

**Keywords:** hepatitis B virus, chronic hepatitis B, antivirals, gene editing

## Abstract

Chronic hepatitis B virus (HBV) infection leads to the development of cirrhosis and hepatocellular carcinoma. Lifelong treatment with nucleotides/nucleoside antiviral agents is effective at suppressing HBV replication, however, adherence to daily therapy can be challenging. This review discusses recent advances in the development of long-acting formulations for HBV treatment and prevention, which could potentially improve adherence. Promising new compounds that target distinct steps of the virus life cycle are summarized. In addition to treatments that suppress viral replication, curative strategies are focused on the elimination of covalently closed circular DNA and the inactivation of the integrated viral DNA from infected hepatocytes. We highlight promising long-acting antivirals and genome editing strategies for the elimination or deactivation of persistent viral DNA products in development.

## 1. Introduction

Hepatitis B virus (HBV) in the human population has existed for 2000–3000 years [1], and the chronic disease affects ~300 million (~4%) people [2,3]. Hepatitis B remains a significant public health issue. It is a leading cause of liver-related deaths worldwide. According to the 2019 World Health Organization (WHO) report, hepatitis B resulted in an estimated 820,000 deaths, mostly from cirrhosis and hepatocellular carcinoma (HCC, primary liver cancer). The highest prevalence of chronic HBV is found in Asia and Africa, whereas its prevalence in the United States (0.3%) is among the lowest and has remained unchanged since 1999 [4]. Use of the HBV vaccine in the USA over the past four decades has significantly reduced the number of acute HBV infections, namely, a 32% reduction was reported from 2019 to 2020 [5]. Now, the vaccination strategy faces two major challenges: documented HBV vaccine protection waning may preclude long-lasting protection [6] migration from endemic regions to Western countries and the USA contributes to stable HBsAg rates despite a predicted steady decline due to universal HBV birth vaccination [7]. These factors influence an increase in HBV-related cirrhosis and HCC mortality since 2010 [8,9].

The WHO has set ambitious targets to eliminate HBV as a public health threat by 2030. HBV “service targets” include further improvement in childhood vaccine coverage (third dose, 82–90%), major increases in birth-dose vaccine coverage or other methods to prevent mother-to-child HBV transmission (38–90%), improved diagnosing (5–90% of people with chronic HBV were diagnosed) and successful treatment (1–80% of eligible people were treated). These service improvements are aligned with “impact targets” of a 90% reduction in new HBV infections and a 65% reduction in HBV-related deaths (when compared to estimates from 2015). Nevertheless, preliminary data suggests that global HBV-related mortality is projected to increase by 39%, from 850,300 deaths in 2015 to 1,109,500 deaths in 2030, if the current diagnosis and treatment timing are not improved [10].

Hepatitis delta virus (HDV) can increase the risk of developing chronic HBV infection as the most severe and progressive form of viral hepatitis in people. The number of HDV-infected persons worldwide ranged from 12 million to 72 million, and therapies to cure hepatitis B could indirectly provide an HDV cure [11]. 

At present, prevention of HBV infection is more appealing since treatment of chronic infection is rarely curative. Advances in HBV pathobiology and the limitations of existing treatments were extensively reviewed and discussed elsewhere [12]. Notably, the progression of chronic HBV to end-stage liver diseases is associated with ongoing viral replication and genomic integration of the portion of the viral genome with oncogenic potential. The major hurdle in the elimination of HBV is associated with the ability of relaxed circular DNA transported to the nucleus of hepatocytes to form stable covalently closed circular HBV DNA (cccDNA), which serves as the template for viral transcription, subsequent replication, and the release of infectious particles, as well as re-entry into the nucleus to replenish cccDNA. Thus, clearance of cccDNA is a fundamental goal for the cure of chronic hepatitis B. Given that all available treatments are unable to eliminate cccDNA from HBV-infected hepatocytes, lifelong therapies are often required. However, very few patients achieve sustained viral and clinical remission from the therapy. In this regard, antiviral therapy with nucleoside/nucleotide analogs (NAs), such as entecavir and tenofovir prodrugs, significantly reduces the incidence of cirrhosis and HCC [13,14,15]. This review summarizes the current and emerging promising therapeutic strategies for chronic HBV treatment and prevention (Figure 1).

## 2. Chronic Hepatitis B (CHB) Outcomes

HBV infects human hepatocytes, and the induced immune response facilitates the elimination of infected cells [16]. Immature immunity or immunosuppression is responsible for the formation of chronic disease or their reactivation after remissions [12,17]. HBV infection in adults mostly results in acute infection (in 90–95% of cases), whereas natal infection results in chronicity in almost all cases, with declining chronicity rates if infection occurs before the age of 5 [18]. Among those living with HBV infection globally, approximately 10% are aware of their status, with an estimated 6.6 million (22%) of those diagnosed undergoing treatment [3]. 

Based on recommendations from the American Association for the Study of Liver Diseases (AASLD), patients with compensated liver disease who are hepatitis B e-antigen (HBeAg)- positive with HBV DNA levels greater than 20,000 IU/mL after a 3- to 6-month period of elevated alanine aminotransferase (ALT) levels greater than two times the upper limit of normal should be considered for antiviral treatment [19]. Compared to acute hepatitis B, which is self-limiting, CHB requires lifelong treatment and consists of different infection phases. These phases include immune tolerant, immune active with HBeAg-positive, inactive, and immune active HBeAg-negative [20]. Positive HBeAg, normal ALT levels, and high levels of HBV DNA, typically above 20,000 IU/mL are the features of the immune tolerant phase. The immune-active phase includes elevated ALT and HBV DNA typically above 2000 IU/mL. In the inactive phase, HBeAg is negative, while hepatitis B e antibody (HBeAb) is positive. ALT is typically normal with undetectable or low levels of HBV DNA (<2000 IU/mL). 

The main goals of antiviral therapy are the suppression of HBV DNA, loss of HBeAg (in patients who were initially HBeAg-positive), and loss of HBsAg. HBsAg seroclearance is a marker of treatment success as it confers favorable clinical outcomes with improved morbidity and mortality [21]. HBsAg seroclearance is achieved by appropriate and timely antiviral therapy and is more likely in patients with less active disease. No association was demonstrated between HBsAg seroclearance and HBV genotype or treatment history. The annual incidence of HBsAg spontaneous seroclearance in patients with chronic HBV is 1.31% [22,23]. 

In Figure 1, we listed clinical forms of chronic HBV and possible outcomes with available treatment. In the absence of treatment with effective suppression of viral replication to undetectable levels and restoration of effective innate and adaptive immune responses, the disease ultimately leads to the development of cirrhosis and a significant risk of HCC. Aging, alcohol misuse, HIV-1 coinfection, malnutrition, or metabolic diseases significantly increase the risk of liver cirrhosis and HCC development [24,25]. Uncontrolled HBV replication in people living with HIV (PLWH) contributes to a higher risk of fibrosis development and mortality [26,27,28].

## 3. Available Therapies to Prevent HBV-Induced End-Stage Liver Diseases

The United States Food and Drug Administration (FDA) has approved eight anti-HBV drugs: two interferons and six nucleoside/nucleotide analogues (NAs).

### 3.1. Interferons

Interferon was the first class of anti-HBV drugs approved by the FDA, with IFN-α being the first drug in the class. Though the exact mode of its action is unknown, different mechanisms have been advanced. One is inhibiting HBV replication by regulating the ccc-driven transcription of pre and subgenomic RNAs, occurring due to cccDNA-bound histone hypoacetylation, co-repressor recruitment, and reduced binding of transcription factors to cccDNA [29]. Additionally, IFN-α possesses immunomodulatory properties that facilitate the activation of natural killer T cells, and B cells, and improve the responsiveness of dendritic cells and monocytes, thereby making it a preferable therapeutic option [30].

A pegylated IFN-α has been commercially available on the market since 2005. It became a first-line treatment due to its advantages, including HBsAg seroclearance and improved dosing regimen [31]. Patients treated with peg-IFN-α monotherapy have been shown to have a lower virological response rate at the end of treatment compared to those treated with NA monotherapy but achieved a higher sustained off-treatment response rate and HBeAg seroconversion rate after treatment-free follow-up for 24–48 weeks [32]. A meta-analysis study showed that the combination therapy of NAs and IFN outperformed IFN monotherapy in both serological and virological responses [33]. However, treatment with interferon has been linked to lower incidences of developing HCC than NAs treatment in patients with high HCC risk [34]. In addition, a recent retrospective study in China has shown the superiority of IFN-α based therapy over NAs in preventing unfavorable outcomes associated with chronic HBV infections such as HCC [35].

Peg-Interferon was the first commercialized long-acting antiviral that replaced standard interferon due to improved shelf-life. Peg-Interferon α-2a (Pegasys^®®^) is manufactured by Genentech, USA, and administrated subcutaneously once a week. Its 40KDa branched-chain PEG conjugate enhances the distribution volume of the drug and prolongs the dosing interval since it cannot be cleared by the kidney [36]. Moreoever, in transgenic mice, IFN-α modified using PASylated technology that involves conjugating drug with Pro, Ala, and/or Ser residues (PAS) showed improved antiviral effects and bioavailability without toxicity when compared to its unmodified form. This provides a proof-of-concept for developing a novel long-acting IFN-α treatment [37].

Nevertheless, patients receiving interferon therapy experienced side effects such as flu-like symptoms, neutropenia, thrombocytopenia, depression, hypo or hyperthyroidism, and skin reactions [38]. 

### 3.2. Nucleoside/Nucleotide Analogous 

NAs act by converting into active triphosphate metabolites, followed by their incorporation into the growing viral DNA chain during replication, which leads to chain termination and inhibition of the viral replication [39]. As of 16 January 2023, the US FDA has approved six NAs for the treatment of HBV, three additional others are approved for use outside the USA, and one is on hold (Hepatitis B Foundation, 2023).

Despite advantages such as an easy oral administration route, improved efficacy and safety profiles, good tolerability, and cost-effectiveness, NAs cause adverse effects such as mitochondrial toxicity, upper respiratory tract infection, nausea, headache, bone mineral loss, and abdominal pain, as reported by [40,41]. 

Multiple studies have examined the effect of NAs treatment on end-stage disease development in HBV patients. For example, a study in China showed that NAs withdrawal could expedite liver failure in chronic HBV patients. Additionally, patients with HBV-related acute-on-chronic liver failure (ACLF) caused by discontinuation of NAs therapy showed a lower spontaneous survival rate in comparison to those with ACLF induced by hepatotropic superinfection [42]. This is supported by the first cohort study, which demonstrated that inappropriate NAs discontinuation led to a high mortality rate in HBV-ACLF patients [43]. Entecavir and tenofovir are recommended first-line NAs agents due to their better mutation resistance profiles. Long treatment with these agents has been shown to reduce HCC incidences in HBV-infected patients [44,45]. In this context, studies demonstrated a lower risk of HCC in patients receiving tenofovir disoproxil fumarate (TDF) compared to those on entecavir therapy [46,47,48,49]. Long-term NAs therapy is associated with sustained suppression of HBV replication that has exhibited improved end-stage diseases conditions such as fibrosis, cirrhosis, and HCC, which were observed in studies involving telbivudine [50,51] and lamivudine [52,53]. Unfortunately, drug resistance remains the major obstacle in this therapy, with lamivudine and adefovir dipivoxil showing high resistance rates—almost 80% and 29%, respectively [54]. On the basis of its established efficacy and tolerability against HIV and HBV, the World Health Organization recommends a tenofovir-containing regimen as the first-line antiviral therapy, particularly, for people living with HIV with HBV coinfection [55]. Antiretroviral therapy containing TDF/emtricitabine (FTC) or TAF/FTC has a prophylactic effect against HBV infection among men who have sex with men with HIV infection in 90% [56]. However, the most important concern with pre-exposure prophylaxis (PrEP) is the improvement of adherence rate, which could be achieved by the development of long-acting agents with anti-HBV activity.

### 3.3. Classes of Anti-HBV Drugs in Development

Novel treatments aimed at inhibiting viral replication/antigen reduction and/or restoring host immune control are in clinical development and have been reviewed in detail by Pan et al. [57]. The most advanced three classes: entry inhibitors, capsid assembly modulators, and immunomodulators are highlighted in this review. Favorable safety and efficacy outcomes will need to be demonstrated for such therapies with novel mechanisms of action to be used in real-life settings.

#### 3.3.1. Entry Inhibitors

Bulevirtide (Myrcludex-B), a promising lead candidate in this class and currently in phase III clinical trials, inhibits HBV infection by competing with HBV particles for the binding site of sodium taurocholate co-transporting polypeptide (NTCP) receptors on hepatocytes [58]. In a Phase I trial, Bulevirtide demonstrated excellent tolerability, and no adverse events were reported [59]. Once every 1–3 days dosing of Bulevirtide was projected in humans based on its liver-targeting ability, exceptional serum stability, and favorable half-life in animal models [60]. In addition to Bulevirtide, other promising entry inhibitor candidates in development include A2342 (Albireo, Boston, MA, USA) and AB-543 (Assembly Biosciences, San Francisco, CA, USA).

#### 3.3.2. Capsid Assembly Modulators

Capsid assembly modulators are one of the recent classes of anti-HBV drugs that inhibit HBV replication by targeting pol-pgRNA encapsidation and blocking early viral life cycle stages, including cccDNA formation [61]. GLS4 (Morphothiadine) and JNJ-56136379 are two lead candidates in this class, currently undergoing Phase II clinical trials. 

Although GLS4 was well tolerated in its first-in-human clinical trial, the concentration required for effective HBV inhibition was not achieved with GLS4 alone, which suggested the need for combination therapy with ritonavir [62]. This combination therapy has further shown antiviral activity and good tolerability in a phase I b trial [63]. 

The combination approach was also efficacious in the NVR 3-778 phase 1 trial, where non-cirrhotic patients showed better HBV DNA and RNA reduction when treated with NVR 3-778 with peg-IFN [64]. Phase I trials of many other capsid assembly modulators are underway.

#### 3.3.3. Immunomodulators

Immunomodulators such as TLR agonists, monoclonal antibodies, checkpoint inhibitors, and therapeutic vaccines can enhance the HBV-specific immune response. Selgantolimod (GS9688, Gilead Sciences, Foster City, CA, USA) was tested in phase 1b study for its TLR-8 agonist activity but failed to demonstrate efficacy [65]. Vesatolimod (GS-9620) demonstrated immunomodulatory activity and safety in phase II trials but exhibited limited effect on HBsAg [66]. Thus far, these studies suggest that the use of TLR agonists alone for treating HBV infections may not improve clinical outcomes.

Although constant efforts have been made in vaccine development, they have yet to prove success in preventing HBV infection. For instance, GS-4774 failed to show any effect on HBsAg in a phase II study [67] even when combined with TDF, which improved CD8+ cell cytokine production but couldn’t reduce HBsAg levels [68,69]. Furthermore, BRII-179 and TG-1050 showed minimal effects on reducing HBsAg levels [70,71].

Treatment with checkpoint inhibitors such as programmed death (PD-1) or ligand (PD-L1) inhibitors have shown increased T cell functions in cancers [72], but their efficiency in treating HBV remains uncertain. Envafolimab, GS 4224, and RG6084 are three inhibitors in clinical trials. However, the use of checkpoint inhibitors in the near future is questionable due to the possible reactivation of HBV, as shown in cancer patients with HBV infection [73].

Overall, results from immunotherapy treatments have been mixed due to challenges, such as lack of standardization in measuring clinical effectiveness, difficulty selecting suitable candidates for therapeutic intervention, and safety concerns [74]. Patients with chronic HBV infection need lifelong treatment, and curative therapies are preferred. 

#### 3.3.4. Gene Therapies

Gene therapy, the delivery of functional nucleic acids or RNPs to specific genetic sites, is now becoming a common point in HBV therapeutic development. A slew of gene therapy strategies has been attempted to stall HBV gene expression through HBV mRNA destabilization/inhibition, host protein modulation, and direct cccDNA mutagenesis. 

RNA interference (RNAi) is currently used to suppress the post-transcriptional functions of HBV RNA transcripts. Here, double-stranded RNA (siRNA) or single-stranded Antisense oligonucleotides (ASO) are designed to contain HBV-specific complementary sequences which either destabilize or facilitate the degradation of viral transcripts upon hybridization [75]. Several RNAi have been developed and modified for improved intracellular delivery, stability, and function. AB-729 is a safe and well-tolerated suppressor of HBsAg and has been reported to provide robust HBV-specific T-cell restoration. In addition, RG-6346 and JNJ-3989 have proven effective in HBsAg reduction during clinical trials [57]. ALG-125755, a subcutaneous N-acetylgalactosamine (GaLNAc) conjugated HBsAg targeting siRNA, is currently in Phase I clinical trial [57]. Due to its 5′ phosphate modification, ALG-125755 demonstrated higher stability and efficacy compared to other siRNAs and is already showing a 71% phase transition success rate. A phase 2b trial was carried out to evaluate the anti-HBV effect of Bepirovirsen (GSK3228836) a 2′-O-methoxyethyl conjugated ASO. The findings from this clinical trial showed sustained loss of HBsAg and HBV DNA following 300 mg Bepirovirsen injection. Clearly, RNAi only transiently silences gene expression and does not provide a complete deactivation of the persistent cccDNA [76]. 

With the growing evidence that cellular host factors (including cyclin D, NQO1, ATM, CHEK 1&2, TOP2A, SMC5/6, and some SIRT family, etc.) play crucial roles in the intrahepatic transcription and replication of HBV, these host factors are being explored as major drug targets to suppress HBV disease progression [77,78,79]. Last year, Wu and colleagues investigated the therapeutic potential of SIRT2 silencing in the treatment of chronic HBV, and their findings were promising. In their studies, they first reported a markedly increased SIRT2 expression in CHB patients compared to healthy individuals. A follow-up SIRT2 gene knockdown in HepG2-NTCP cells and HBV-infected PHH significantly reduced the levels of viral RNAs, HBsAg and HBeAg. Wang et al. [79] first discovered a small molecule compound that appears to be a game changer in the quest for an HBV cure. The novel compound demonstrated potent anti-cccDNA activity in highly relevant natural HBV infection models. The authors reported significant inhibition of HBV DNA, HBsAg, and HBeAg. Of note, the compound also reduced pgRNA, thus preventing the cccDNA intrahepatic replenishment while eliminating the already established cccDNA pool. Although the mechanism of action of this compound remains unclear, its interaction with cellular-host factors and induction of cytokines has been strongly implicated.

The stability of cccDNA makes the permanent eradication of HBV extremely difficult, and new gene therapies directly targeting cccDNA are currently underway. Currently, DNA nucleases and their modern variants are being utilized to induce cleavage and single base edits in the HBV genome, but their large sizes and off-target effects are a major concern (discussed in Section 5). A relatively small (~1.1 kb) polymerase-targeting engineered nuclease, ARCUS-Pol, was developed by Gorsuch et al. [80]. ARCUS-Pol mRNA was readily encapsulated in lipid nanoparticles and bioavailable in HBV-infected primary human hepatocytes and non-human primates. The authors reported that indels generated by the engineered nuclease led to an 85%, 80% and 77% reduction in cccDNA, extracellular HBV DNA and HBsAg, respectively. Despite these promising findings, the induction of off-target double-stranded breaks is undesirable.

#### 3.3.5. Small Molecule Inhibitors

Unlike nucleos(t)ide therapies, which reduce viremia but do not lead to an effective reduction in HBV antigen expression, new small molecules (such as RG7834 [81], ccc_R08 [79] and several others [82] significantly reduced the levels of viral proteins (including HBsAg), as well as lowering viremia. The mechanisms of action of these molecules are not clear and are mostly related to interference with viral RNA, HBsAg production, and host factors involved in HBV replication and secretion, as mentioned in Section 3.3.4. RG7834 is a small molecule compound belonging to the chemical class of the dihydroquinolizinones that was discovered by phenotypic screening for inhibitors of HBsAg secretion from a library containing approximately one million small molecule compounds. RG7834 demonstrated a higher degree of selectivity for sub-genomic RNAs. Unfortunately, in vivo, efficacy in the humanized liver model was limited and oral doses of 4 and 10 mg/kg twice daily for 21 days only slightly reduced HBV DNA, HBsAg and HBeAg in blood, and did not affect liver tissue viral markers. However, the combinations of both RG7834 and ETV, and RG7834 and PegIFNα led to profound and significant reduction 5 in both HBV DNA and HBsAg levels in the blood, but not in tissue levels of cccDNA. Another group of tetrahydropyridine derivatives showed efficacy in HBV transgenic mice comparable with RG7834 [82]. 

Another small molecule is a flavonoid derivative ccc_R08, a specific HBV cccDNA inhibitor [79]. This compound in vitro on human hepatocytes dose-dependently inhibited not only extracellular HBsAg but also extracellular HBeAg and DNA as well as intracellular viral DNA and RNA, with IC50 values ranging from 0.2 to 5 μM. Treatment of HBV cccDNA+ mice (HBVcircle model) twice a day with 20 mg/kg for two weeks not only completely suppressed HBV markers in peripheral blood to undetectable levels, but also eliminated cccDNA from liver tissue. This effect prevents viral rebound. Since ccc_R08 was found to be associated with two safety flags: CYP1A2 induction and phototoxicity, a close homolog ccc_R09 was tested on humanized liver mouse model (uPA-SCID) at a dose of 200 mg/kg for four weeks and showed low efficacy. The differences in activity could be related to the model used. If in the HBVcircle model, there was no re-infection mouse hepatocytes step, human hepatocytes in uPA-SCID mice were constantly re-infected making complete elimination of cccDNA questionable. The investigation of the mechanisms of action of ccc_R08 provides the ground to generate new knowledge on the cellular networks involved in the persistence and regulation of the HBV minichromosome [83]. 

## 4. Long-Acting Therapeutics

Conventional oral drug therapy provides a typical rapid pulse release and absorption pattern, leading to rapid fluctuation in systemic drug concentration. Poor adherence to treatment or missed doses can exacerbate the disease condition. Long-acting (LA) therapies can provide favorable pharmacokinetics and pharmacodynamic profiles of the drug which allows for the extension of dosing intervals from hours to weeks or months. Less frequent dosing can significantly improve the treatment outcomes in patient subpopulations who struggle with daily dosing and prefer longer-acting treatment interventions. 

Long-acting modalities rely on in vivo formation of a drug-eluting “depot” to achieve sustained drug release over an extended period of time. Recent advances in biomaterial and medicinal chemistry have led to innovations that have allowed for extended dosing schedules for various therapeutics, such as contraceptives [84], anti-psychotics [85,86], and HIV-1 therapies. 

For antiviral therapies, remarkable progress has been made in creating long-acting formulations and devices for HIV treatment and prevention [87]. Notable approvals include Sunlenca [88], Cabenuva [89], and Apretude [90]. However, none of these agents are active against a high-risk HBV infection in HIV patients. The HPTN 084 phase 3 randomized clinical trial provides evidence that a long-acting Apretude is superior to conventional daily oral Truvada in preventing HIV infections in men who have sex with men and transgender women, underscoring the significance for the development of long-acting agents for overlapping disease conditions [91]. Long-acting injectable (LAI) antiretroviral therapy (ART) has simplified regimens and could significantly improve adherence [92].

### 4.1. Long-Acting Injectables for Chronic HBV Treatments

The development of LAI for chronic viral infections beyond HIV, including HBV, is gaining traction due to its potential to simplify regimens and treatment outcomes. Encouraging preclinical research has been conducted on LAI formulations of emtricitabine, lamivudine, tenofovir, and entecavir [93,94,95,96,97,98,99,100,101]. LAI anti-HBV agents can be particularly beneficial for high-risk patients facing treatment adherence and stigma challenges. Furthermore, surveys and clinical trials in people living with HIV demonstrated a greater preference for LAI therapeutics [102,103]. The need for LAI treatments for chronic diseases has continued to grow due to improved safety, tolerability, and efficacy. LAI treatment may be considered for preventing HBV reactivations in patients undergoing immunosuppressive therapies or even hepatitis C treatment since HBV rebounds have been reported in these settings (especially in HBsAg+ but also in HBsAg-neg/anti-HBc+ individuals) [104,105,106]. The use of ultra-long-acting antivirals for the prevention of certain viral illnesses, halting either contagions or reactivations under immunosuppression would fill an immediate need in providing protection when classic vaccines do not exist, responses are suboptimal, escape mutants emerge or immunity wanes [107].

Antiretroviral drugs belonging to the class of nucleoside reverse transcriptase inhibitors (NRTI) show activity against HBV. These are tenofovir alafenamide (TAF), tenofovir disoproxil (TDF), entecavir (ETV), emtricitabine (FTC), lamivudine (3TC), adefovir and telbivudine [108]. Of these, tenofovir and entecavir are preferentially recommended for first and second-line treatment compared to other NAs due to their higher potency and barrier to resistance [19,109,110].

To address adherence to long-term chronic HBV treatment, the frequency of once daily dosage must be decreased to once a week or longer. Currently, there are no approved long-acting antiretroviral drugs for the treatment or prevention of HBV infection. However, several pre-clinical studies using various drug delivery approaches have demonstrated the potential for extending the half-lives of NAs used to manage HBV and HIV infections. 

One method for developing long acting injectables is through the application of prodrug strategies to existing antiviral therapies. Lipophilic long-acting aqueous stable ester prodrugs and ProTides of TFV, FTC, and 3TC have been reported with promising preclinical pharmacokinetics and efficacy results [93,95,97,111], and 3TC [98,99,101]. Notably, two modified ProTides of TFV were synthesized and nanoformulated into surfactant-stabilized aqueous nanocrystals. A single intramuscular injection of these modified TFV ProTide nanocrystals to Sprague Dawley rats at 75 mg/kg TFV equivalents produced sustained efficacious TFV-DP levels over two months in PBMCs and other key HIV and HBV target tissues and cells [93]. The lead NM1TFV ProTide formulation was evaluated in HBV-infected transgenic and human hepatocyte transplanted TK-NOG mice and exhibited sustained antiviral activities for at least three months in both models, while an equivalent dose of a nanoformulated TAF control formulation (NTAF) had limited effect on suppression of HBV DNA replication [111]. The antiviral activities were also studied in HBV-transduced hepatoma cells where NM1TFV was shown to stimulate innate immunity genes (type I IFN and *ISG15*) expression. The pharmacokinetic and pharmacodynamic studies of M1TFV showed no adverse reaction on the injection site [93]. 

A similar long-acting lamivudine ProTide named NM23TC was shown to sustain drug levels in blood and tissues for at least 30 days after a single intramuscular injection in Sprague Dawley rats [99]. This prodrug was further tested in an HBV-infected chimeric liver mouse model where a single intramuscular injection of NM23TC at 75 mg/kg 3TC equivalents suppressed HBV replication for 4 weeks [98]. The modified ProTide approach has also been used to extend the apparent half-life of emtricitabine, where sustained prodrug and active drug (FTC-TP) levels were recorded in peripheral blood mononuclear cells (PBMCs) and tissues over a period of one month following a single intramuscular injection of the nanoformulated FTC ProTide at 45 mg/kg FTC equivalents to Sprague Dawley rats [95].

Elsewhere, Ho et al. synthesized a lipidic ester prodrug of ETV called entecavir-3-palmitate (palmitate ester prodrug of entecavir) and formulated it into aqueous microparticles using the anti-solvent crystallization method. In vivo, pharmacokinetic evaluations in rats showed sustained plasma ETV concentrations for up to four weeks following a single subcutaneous injection of 1.44mg/kg of ETV. The microparticles were well tolerated at higher doses [94]. In a separate study by Zhang et al., ETV was formulated as an extended-release poly (lactic-co-glycolic acid) microsphere and exhibited plasma ETV levels for 42 days in rats following a single intramuscular injection at 1.5 mg/kg. In vitro, the ETV microsphere formulation demonstrated a sustained release for about two months. [112]. Further drug biodistribution, active metabolite quantitation and HBV efficacy studies in preclinical models of disease will potentially provide more insights into correlations between plasma ETV levels and HBV efficacy.

Another approach that could potentially be used to manage HIV-HBV co-infections, if therapeutic drug levels are delivered to sites of viral replication including liver hepatocytes for HBV infection, is the application of lipid nanoparticles (LNPs) loaded with multiple antiretrovirals. Freeling et al. showed that LNPs loaded with a combination of TFV, lopinavir (LPV), and ritonavir (RTV) achieve sustained drug levels in plasma, PBMCs and lymph nodes for 7 days after a single subcutaneous dose of LPV, RTV, and TFV at 25.0, 14.3, and 17.1 mg/kg, respectively in macaques [113]. Formulation optimization further improved the PK profile of the loaded drugs extending their half-lives by an additional week in rhesus macaques [114]. A similar lipid formulation loaded with LPV, RTV, TFV, and 3TC has been reported where drug levels were sustained for 5 weeks following a single subcutaneous injection of 25.0, 7.0, 10.6, and 10.6 mg/kg of LPV, RTV, TFV, and 3TC, respectively in rhesus macaques [100]. Given HBV infection and replication takes place in the liver, drug delivery into hepatocytes using this approach will be critical to control infection. 

### 4.2. Implants

Implantable drug delivery devices and injectables generally consist of a drug reservoir surrounded by a polymer matrix or a drug-polymer mixture. Multiple kinds of implants have been developed in the field of long-acting antiretroviral drug delivery, of which Tenofovir- and Entecavir-loaded implants find relevance, in the field of HBV treatment even though none of them have been evaluated for HBV efficacy.

#### 4.2.1. TAF Implants

A biodegradable, tunable, subcutaneous implant of TAF was developed for HIV PrEP. This thin-film polymer device showed an in vitro linear release of TAF at 1.2 mg/day for up to 90 days and 2.2 mg/day for up to 60 days [115]. Another reservoir-style, tunable implant filled with a formulation of TAF and castor oil excipient showed sustained release of TAF at 0.28 ± 0.06 mg/day over the course of 180 days in vitro [116]. Further optimization of the biodegradable subcutaneous implant strategy led to the development of a TAF freebase platform that maintains the high chemical stability of the prodrug for up to 240 days of in-vitro exposure [117].

Chua et al. developed a transcutaneous refillable TAF and FTC implant for HIV PrEP. In rhesus macaques, this nanochannel delivery implant demonstrated varying but sustained release of both TAF and FTC in plasma for 83 days as well as effective drug distribution to tissues relevant to transmission. Preventive levels of tenofovir-diphosphate were achieved as early as 3 days after implantation and were sustained over the duration of the experiment. However, preventive levels of FTC triphosphate were not achieved, likely, due to the high dosage of FTC required [118]. A non-blinded, placebo-controlled rhesus macaque efficacy assessment of the subcutaneous TAF nanofluidic implant showed a 62.50% infection risk reduction compared to control animals after repeated low-dose rectal SHIV_SF162P3_ challenges. The pharmacokinetic profile showed preventive TFV-DP PBMC concentrations one-day post-implantation and sustained at a median of 390.00 fmol/10^6^ cells for 4 months. Immunohistochemistry of the tissue surrounding the implants showed fibrotic capsules with limited cellular infiltration. The implants were well tolerated, with no notable significant differences observed between the treated and placebo groups [119].

A long-acting, subcutaneous, non-biodegradable reservoir-type, rod-shaped implant filled with TAF hemifumarate microtablets was described for HIV prevention. In vitro drug release was sustained for 100 days [120]. In vivo, studies showed sustained drug levels for over 90 days. However, a dose-ranging study in New Zealand white rabbits showed severe necrosis around the active implant sites. On lowering the rate and flux of TAF in rhesus macaque, local inflammation and necrosis near the implants were still observed. Histological inflammatory response at 4 and 12 weeks was scored severe in rhesus macaques [121]. 

In another study, Gunawardana et al., developed a long-acting, non-biodegradable, silicone subdermal implant of TAF. In vivo, pharmacokinetic evaluations in beagle dogs showed sustained release of TAF for 40 days. No treatment-related adverse events were observed in the animals [122]. A 6-month safety and pharmacokinetics study in mice and sheep demonstrated similar drug release kinetics with no clinical safety concerns apart from the expected foreign body response [123]. This device is currently under evaluation for safety, acceptability, tolerability, and pharmacokinetics in phase I/II clinical trials (CAPRISA 018) [124].

#### 4.2.2. Entecavir Implant

Two types of subcutaneous implants of Entecavir were created entecavir-poly(caprolactone) hot melt extrudates and compressed entecavir dip-coated tablets. In Wistar rats, entecavir-polymer extrudates demonstrated sub-efficacious drug release for over 180 days. Local cutaneous swelling, scab formation, and extensive necrosis were observed around the implant area, which progressed to the expulsion of the drug-loaded implant from the skin. This led to the conclusion that entecavir was not tolerated at the implant site at drug release rates estimated to be efficacious for HBV treatment [125].

#### 4.2.3. Long-Acting Orals, Patches and Rings

Advances in medicinal chemistry led to the identification of a potent long-acting nucleoside reverse transcriptase inhibitor E-CFCP by Higashi–Kuwata. The inhibitor exhibited potent therapeutic efficacy against wild-type HBV and drug-resistant variants in vitro. Studies in HBV-infected human-liver-chimeric mice suggest the potential for once-weekly oral dosing of E-CFCP [126].

Long-acting transdermal patches have been explored for long-acting drug delivery. Puri et al. prepared two kinds of patches: transparent patches using acrylate adhesive and suspension patches using silicone and polyisobutylene adhesives. In vitro permeation studies from the silicone-based suspension patch released TAF at the target release profile of 8.4 mg/day projected for weekly drug dosing [127]. Pharmacokinetic evaluation of the silicone-based patch in female hairless rats achieved sustained drug levels in plasma and vaginal tissue. Skin adhesion and tolerance of the TAF transdermal patch were also acceptable [128].

Kiser and colleagues designed a long-acting intravaginal ring of tenofovir by filling polyurethane tubing of various hydrophilicity with a high-density TFV/glycerol/water semisolid paste. In vivo pharmacokinetic studies in sheep showed a released profile of 10-mg/day TFV maintaining drug levels at approximately 10^4^ ng/g, 10^6^ ng/g, and 10^1^ ng/mL over 90 days, in the vaginal tissue, vaginal fluid, and plasma, respectively [129]. This TFV intravaginal ring was further modified to incorporate levonorgestrel (hormonal birth control) by induction welding the separate drug eluting segments. Levonorgestrel was also shown to sustain dose-dependent levels in plasma and cervical tissue for 90 days [130]. This long-acting intravaginal ring delivering tenofovir at 10 mg/day alone or with levonorgestrel at 20 mg/day for 90 days was studied in a randomized, placebo-controlled, phase 1 study- CONRAD A13-128.

The ring was found to be safe and well-tolerated. Participants randomized to TFV-containing vaginal rings showed cervical aspirate TFV concentrations exceeding 1000 ng/mL within 4 h of insertion and an average of 1814 fmol/mg TFV-DP concentrations in the vaginal tissues 72 h post removal. TFV/levonorgestrel intravaginal ring group showed higher anovulation rates, cervical mucus alterations, and abnormal sperm penetration compared to controls. This ring is further being tested for its intended duration of use in an extended 90-day safety, PK and PD study in women [CONRAD A15-138 study] [131].

Exciting progress has been made in developing LA therapies for treating HBV infection, and these drugs hold great promise for the future of HBV treatment, including long-acting TFV for the prevention of mother-to-child transmission [132,133,134,135].

## 5. Gene Editing Manipulations to Disrupt HBV cccDNA and Integrated Viral Genes

The ideal goal of antiviral therapy is to achieve a cure, which largely depends on suppressing the pool of cccDNA and HBsAg. HBV cure strategy can be broadly categorized into two types: functional cure and sterile cure. A functional cure for HBV involves the loss of HBV surface antigens (HBsAg), with or without the development of antibodies against HBsAg (anti-HBs), as well as undetectable serum DNA and the persistence of cccDNA with low or no transcriptional activity, allowing for treatment to be stopped. In a functional cure, the activated immune system should be able to control the remaining infected cells. However, a sterile cure of HBV would require the complete elimination of the cccDNA [136,137].

### 5.1. Application of Gene Editing Technology in Chronic Hepatitis B Infection

The recent advancement in technology has led to the development of new gene editing tools capable of mutagenizing or completely degrading the long-lived forms of HBV DNA [138,139,140]. The persistent expression of viral proteins from the pre-genomic transcripts of the long-lived episomal DNA (cccDNA) and chromosomally integrated HBV DNA is the hallmark of chronic hepatitis B disease progression and a major obstacle to achieving a functional cure [141]. The goal of developing new therapeutics for CHB has shifted from managing HBV viremia and its concomitant sequel to achieving a functional cure defined by the permanent loss of HBsAg and HBV DNA [142]. This requires curative strategies that specifically target and disrupt the cccDNA and HBV DNA integrant responsible for the persistence of HBsAg and HBV DNA after treatment with the currently approved therapies. The cccDNA is a fairly small (~3.2 kb), compact genome containing tightly overlapped open reading frames (ORFs) encoding the genes necessary for viral survival. These unique qualities, in addition to its few copy numbers per infected cell, make cccDNA a potential substrate for gene editing. 

Gene editing systems utilize engineered DNA-binding nucleases to generate site-specific insertion-deletion (indels) mutations. Their mode of action has advanced from induction of double-stranded break followed by spontaneous DNA repair to a less invasive and more precise single nucleotide modification [143,144]. Meganucleases, transcription activator-like effector nucleases (TALENs), and zinc finger nucleases (ZFNs) were the earliest gene editing systems explored by scientists to provide a proof of concept that essential genetic sequences in the cccDNA are accessible for functional cleavage [145,146,147]. However, their clinical development and translation have proved abortive due to a lack of specificity, laborious and expensive optimization requirements, and poor packaging [148]. 

Today, these setbacks have been largely resolved using cheap, efficient, and highly adaptable engineered RNA-guided CRISPR systems. Originally identified as a bacteria’s immune defense system, CRISPR technology is now widely recognized as a promising tool to permanently eliminate HBV expression with high specificity. Generally, the wild-type (WT) CRISPR system is a ribonucleoprotein (RNP) complex that comprises a nuclease enzyme (Cas) joined to a user-defined guide RNA (gRNA) by a tracRNA (trans-activating CRISPR RNA). Although more permissive variants have been developed, the WT CRISPR gene editing activity is restricted to genetic sequences that contain Protospacer Adjacent Motif (PAM) and this motif sequence varies between all Cas nuclease subtypes [149]. The presence of a PAM sequence activates a cascade of DNA modifying events in the following succession: target sequence recognition, DNA helix unwinding, double-stranded DNA cleavage and repair through the non-homologous end-joining pathway (NHEJ). The sequence-specific cleavage activity of WT Cas begins with the assembly of the entire CRISPR machinery at the PAM region flanking the target site in the HBV genome under the guidance of the gRNA. The HBV DNA helix then unwinds upon Cas binding to the PAM sequence allowing the gRNA to hybridize with its complementary DNA. When a perfect complementarity is achieved, the WT Cas nuclease is activated to cleave both strands of the DNA and allowed to repair through an error-prone NHEJ [144]. Several studies have leveraged the specificity of the WT CRISPR/Cas system to block the expression of multiple HBV genes using single or multiplexed gRNAs in clinically relevant in vitro and in vivo models with very encouraging results [139,150]. However, off-target effects still exist.

In gene-based therapy, the objective is to harness the specificity of an RNA-guided CRISPR system to permanently inactivate persistent HBV DNA forms without editing the host genome or creating new active viral species. The WT Cas may induce double-stranded breaks (DSB) in the host genome resulting in a lethal frameshift or chromosomal recombination following the error-prone NHEJ repair [149,151]. Also, accumulating lines of evidence suggest that cleavage or linearization of cccDNA may facilitate their integration into the host genome, as frequently observed in chronic hepatitis infections, thus jeopardizing their therapeutic effect [152,153]. Additionally, in a thorough investigation of the fate of cccDNA following WT Cas cleavage, Martinez and colleagues reported the emergence of new transcriptionally active cccDNA variants [139]. 

Less invasive Cas-derived base editors and prime editors capable of creating single nucleotide alterations without inducing DSBs were created to address these concerns [144]. Base editors (BEs) can introduce a permanent point mutation in a DNA sequence through the chemical deamination of single nucleobases. Generally, BEs are classified into cytosine base editors (CBEs) and adenine base editors (ABEs), and both BEs are rapidly evolving into more efficient and specific designer nucleases [144]. Currently, CBEs is a trimeric protein complex consisting of an APOBEC deaminase enzyme, a Uracil Glycosylase Inhibitor (UGI), and a Cas9 nickase [154]. The APOBEC was adopted in CBEs design due to its natural antiviral role during acute HBV infection. When induced by IFN-α, APOBEC attacks foreign DNA, catalyzing the deamination of cytosine to uracil, which is then converted to thymine by a mismatch repair system [155]. To achieve a programmable and precision C/G—T/A transition in a modern base editing system, the cytidine deaminase first converts the cytosine within the editing window to uracil. The UGI then protects the newly formed uracil from rapid reversal because DNA uracil bases are rapidly eliminated by the cell’s Uracil DNA Glycosylase (UDG) enzyme for base excision repair initiation. Finally, a Cas9 variant possessing a nickase activity (Cas9n), nicks the single strand containing the complementary guanine as a signal for excision and correction to adenine through the base excision repair pathway [144].

### 5.2. Inactivation of Persistent HBV DNA Forms Using Base Editors

Yan and his team first discovered rewriting the genetic code of episomal cccDNA into a dysfunctional phenotype using CBEs. They successfully introduced premature stop codons in the overlapping regions of polymerase and surface genes, which significantly decreased the expression of HBsAg and HBV polymerase in naturally infected HEPG2-NTCP cells [156]. In theory, because the HBsAg secretion in chronic HBV infection largely originates from HBV DNA integrants in the host genome, mutagenizing the highly conserved domains in the S gene could potentially suppress the expression of HBsAg. To test this hypothesis, Zhou et al. investigated the effect of CBE-induced point mutations on the 30th codon of the S gene using Integrated HBV DNA substrates in PLC/PRF/5 hepatoma cells. Lentiviral expression of CBEs in PLC/PRF/5 reduced extracellular and intracellular HBsAg by 92% and 84%, respectively [153]. 

An even more relevant and promising finding was presented at the recently concluded HBV 2022 International Meeting. In this elegant study, LNP-delivered CBE mRNA led to a sustained loss of key viral proteins (HBsAg and HBeAg) and HBV DNA resulting in an impeded viral rebound in lamivudine-pretreated primary human hepatocyte (PHH) and mice harboring cccDNA surrogate compared to only lamivudine treated groups [157]. The robust bioavailability and gene editing activity of the transiently expressed CBEs utilized in this study represents an important milestone in developing safe therapeutic BEs as it addresses two critical issues: safe delivery vehicles and the regulation of BE activities in the cells. Following the approval of Onpattro (an LNP-based RNAi drug that treats hereditary transthyretin-mediated amyloidosis) and the most recent mRNA COVID vaccines, LNPs have become widely utilized for the delivery of nucleic acids, including mRNAs, into target cells because of their advantages over viral vectors [158,159]. LNPs readily entrap RNA payloads into their interior space, conferring systemic protection against enzymatic degradation. Furthermore, their rapid diffusion coefficient across the cell membrane aids in the transport of mRNA into the cytosol, where they are translated into a functional protein [160]. Importantly, LNPs are safe for drug delivery purposes as they do not possess immunogenic viral proteins [161]. Aside from the need for safe delivery vehicles, the regulation of BE expression in the host cell must also be carefully considered. The constitutive expression of BEs and gRNAs through viral vectors is hazardous to the human host as part or all of these foreign DNA fragments may undergo random integration into the host genome leading to increased off-target effects and hepatocellular carcinoma [162]. The transient expression of active BEs components as mRNA or ready-to-use protein forms enhances gene editing efficiency and rapid clearance, thereby reducing unwanted chromosomal integration of BE DNA and off-target effects [163] (Figure 2). 

It has been previously reported that cell-derived nanoparticles called exosomes can successfully package and deliver cas9 RNPs to HBV-expressing cells where they demonstrate significant DSB-induced viral gene knockdown [164]. Wang et al. demonstrated that exosomes secreted by cas9/gRNA expressing Huh7 cells contained gRNA and Cas9 protein which showed HBV-specific gene editing following exosomes uptake in HBV-expressing Huh7 cells. This interesting finding suggests that exosomes may be used to deliver BE and gRNA RNPs to HBV-infected hepatocytes. However, it is crucial to select safe cell sources with scalable exosome secretion, high hepatocyte targeting profile, controlled BE loading and release, and minimal immunogenicity. Future gene therapy studies should focus on the optimization of safe and highly efficient BE variants with little or no off-target mutations, targeted non-viral delivery systems with efficient BE packaging and release, and utilization of naturally infected humanized mice models and non-human primates. 

## 6. Summary

The prevention of end-stage liver diseases associated with chronic HBV is based on antiviral therapy with the goal of achieving a cure, which largely depends on suppressing the pool of cccDNA and HBsAg. HBV cure strategy can be broadly categorized into two types: functional cure and sterile cure. A functional cure for HBV involves the loss of HBsAg, with or without the development of antibodies against HBsAg, as well as undetectable serum DNA and the persistence of cccDNA with low or no transcriptional activity, making the treatment able to be stopped. In a functional cure, the activated immune system should be able to control the remaining infected cells. However, a sterile cure of HBV would require the complete elimination of the cccDNA [136,137].

Achievement of HBV functional cure focuses on eliminating HBsAg and suppression/elimination of cccDNA. Approved IFN/PEG-IFN therapy mainly has immunostimulatory effects and can induce HBsAg negativity, but the success rate is low and adverse events are common. Alternatively, HBsAg secretory inhibitor can also be used to prevent the secretion of HBsAg. Considering HBsAg can also use canonical host-cell secretory pathways, this strategy may also be unsuccessful. Nevertheless, Rep2139, an inhibitor of HBsAg secretion, when combined with PEG-IFN-α induced a significant decline in viremia and seroconversion in favorable HBsAg responders [165,166].

NAs aim to inhibit HBV DNA replication and reduce HBV DNA and ALT levels but rarely reduce cccDNA pool in the liver. Although currently approved NAs cannot alone achieve a functional cure, the chances of achieving a long-lasting functional cure for hepatitis B, with no rebound after stopping therapy, would be higher using a combination of immunomodulators, such as IFN-α and two or more antivirals that can target different steps in the virus replication cycle. A combination of virus entry blockers and maturation inhibitors, such as oral CAMs can be very effective in this regard [167]. A combination of NAs and CAMs can also be very effective because of their ability to target critical steps in the HBV replication cycle [168], as shown in Figure 2.

When patients need treatments, the liver cccDNA pool is already established. Due to its long half-life and in the absence of massive liver regeneration, it has been mathematically modeled that long-term treatment with NAs would be necessary to significantly impact this pool [146,147]. The use of long-acting antiviral drugs can be advantageous as patients will have a steady state plasma concentration of the antiviral drugs, allowing for easier adherence. Achieving a functional cure, such as undetectable serum HBV-DNA plus HBsAg loss, is associated with excellent long-term outcomes [168]. Preclinical studies demonstrated that LA medicine is more effective for long-term viral suppression compared to oral medication [98,111]. Long-term viral suppression using LA formulations can also provide a predictable landscape of cccDNA (Figure 2).

## 7. Conclusions

The road to achieving a higher functional cure rate for HBV and prevention of end-stage liver diseases is challenging. It is apparent that neither antiviral medications nor immunomodulators can achieve a functional cure on their own, indicating that further improvements are necessary to increase the likelihood of a functional cure. These improvements may include developing long-acting medication to provide sustained viral suppression and improve the chances of effectively reducing HBsAg level and silencing the cccDNA.

## Figures and Tables

**Figure 1 biomolecules-13-01208-f001:**
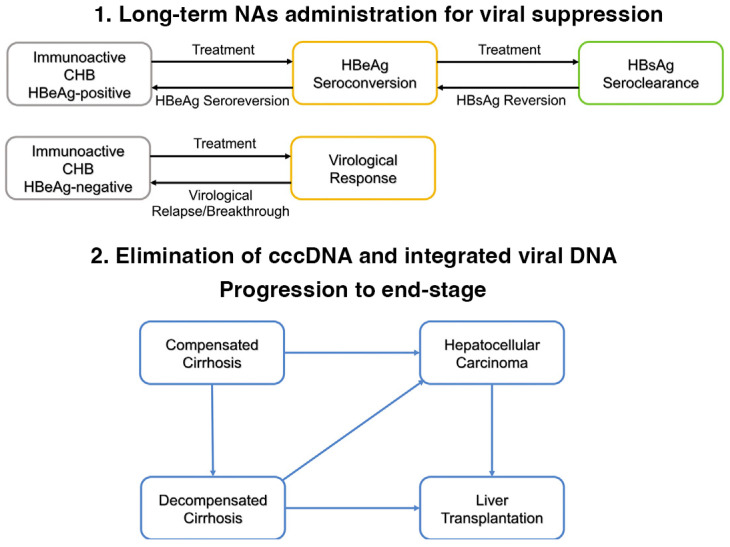
Therapeutic interventions for the prevention of end-stage liver diseases by effective suppression of HBV replication (1) and elimination of viral DNA products (2).

**Figure 2 biomolecules-13-01208-f002:**
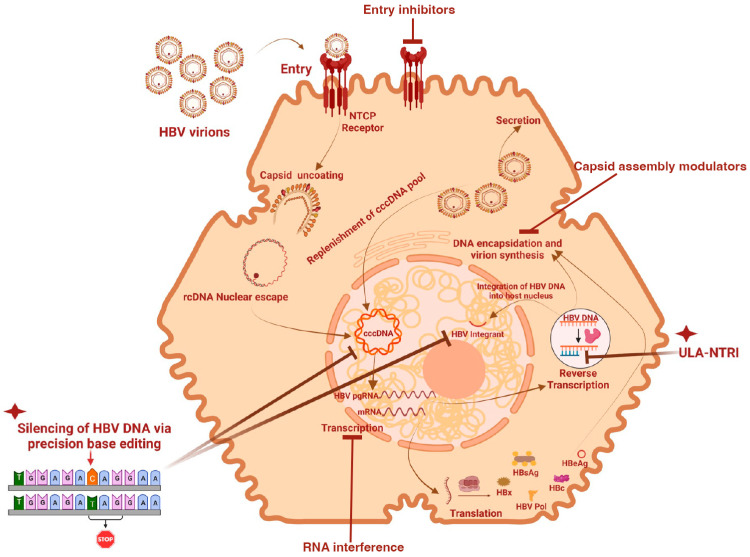
Current therapeutic strategies for HBV treatment and cure. Entry inhibitors block the NTCP receptors preventing viral entry into hepatocytes; Base editing tools selectively rewrite persistent HBV cccDNA and integrants into dysfunctional phenotype; RNA interference targets HBV RNA to prevent its translation into viral proteins; Capsid assembly modulators block the encapsidation Pol-pgRNA and cccDNA replenishment; and Ultralong-acting Nucleotide Reverse Transcriptase Inhibitors (ULA-NTRI) directly inhibit reverse transcription of HBV RNA, thereby providing sustained suppression of HBV DNA synthesis and chromosomal DNA integration. Figure created with BioRender.

## Data Availability

Data sharing is not applicable. No new data were created or analyzed in this study. Data sharing is not relevant to this article.

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
