# Peer review of "Chronic Hepatitis B Infection: New Approaches towards Cure"

_biomolecules, 2023, doi:10.3390/biom13081208_

Round 1
Reviewer 1 Report
he manuscript is a detailed and quite comprehensive review of all the strategies of treatment of chronic HBV infection which is a significant public health problem. The chronic HVB infection is still a life-threatening disease requiring now life-long treatment. The existing treatment strategies are often not sufficient because of appearing the drug-resistant strains or significant side effects. The authors are the first who included into the review all the potentially perspective strategies to remove from the human organism DNA copies of HBV genome. The reviewer did not find significant mistakes or gaps in this review. It deserved the publication.
Author Response
Thank you very much for positive comments.
Reviewer 2 Report
This is a nice and updated review on the newest therapeutic agents on development to confront hepatitis B virus infection. Although the overall scope of new agents and strategies is covered by the authors, the manuscript would benefit form some further clinical touch.
Specific comments:
1. Title. Consider to be more accurate. The new agents pursue the treatment of HBV infection and not just preventing end-stage liver disease. Therefore, consider something like 'Chronic hepatitis B virus infection: new approaches towards cure'.
2. Gene therapies other than gene editing tools should be mentioned and discussed briefly. Consider re-phrasing 3.3: ‘Classes of anti-HBV’, adding a new section 3.3.4 after Immunomodulators.
3. The authors should discuss the impact of oral small-molecules such as ccc_R08 that disrupt cccDNA (check and refers to: Wang et al. Discovery of a first-in class orally available HBV cccDNA inhibitor. J Hepatol 2023; 78: 742–53; Zoulim et al. Eliminating cccDNA to cure hepatitis B virus infection. J Hepatol 2023; 78: 677-80.
4. Hepatitis delta affects roughly 5% of chronic hepatitis B patients globally. A short section discussing the challenges of this subset of patients would be worth to be included. The good news is that the achievement of functional HBV cure or, even better, eliminating/silencing HBV cccDNA could provide HDV cure indirectly. Check and refers to: Hepatitis D virus infection. N Engl J Med. 2023; 389: 58-70.
5. Although the authors state that HBV prevention expanding HBV vaccines would be the more efficacious way to reduce global HBV prevalence, two major challenges should be acknowledged. Firstly, HBV vaccine protection waning has been documented and may preclude long-lasting protection. In this regard, unexpected high rates of lack of HBV seroprotection have been found in recent surveys (i.e., Soriano et al. Susceptibility to hepatitis B virus infection in adults living in Spain. Liver Int. 2023; 43: 1015-20). Secondly, migration flows to Western countries (North America and EU) from HBV endemic regions in Asia and Africa contribute to keep stable HBsAg rates precluding the steadily declines expected because of universal HBV birth vaccination.
6. Long-acting formulations of anti-HBV agents may be considered for indications other than treating chronic hepatitis B. This is the case for preventing HBV reactivations in patients undergoing immunosuppressive therapies or even hepatitis C treatment, since HBV rebounds have been reported in these settings (especially in HBsAg+ but also in HBsAg-neg/anti-HBc+ individuals). Consider adding a paragraph discussing this topic at the end of section 4.2. Check and refers to: Soriano et al. Ultra-long-acting antivirals as chemical vaccines to prevent viral diseases. Future Microbiol. 2022; 17: 887-97.
English grammar is adequate and only needs minor revision and editing.
Author Response
We want to thank all reviewers for spending time reviewing the papers, and special thanks to reviewer 2 for his great and helpful suggestions.
We are pleased that reviewers characterized this paper as “a nice and updated review on the newest therapeutic agents on development to confront hepatitis B virus infection”. As they suggested, “although the overall scope of new agents and strategies is covered by the authors, the manuscript would benefit from some further clinical touch”. Thus, we made the requested changes.
Specific comments:
- Consider to be more accurate. The new agents pursue the treatment of HBV infection and not just preventing end-stage liver disease. Therefore, consider something like 'Chronic hepatitis B virus infection: new approaches towards cure'.
Response. Thank you for this great suggestion. We changed the title as recommended.
- Gene therapies other than gene editing tools should be mentioned and discussed briefly. Consider re-phrasing 3.3: ‘Classes of anti-HBV’, adding a new section 3.3.4 after Immunomodulators.
Response. As per the request of the reviewer, we included a new section 3.3.4 discussing gene therapies.
- The authors should discuss the impact of oral small-molecules such as ccc_R08 that disrupt cccDNA (check and refers to: Wang et al. Discovery of a first-in class orally available HBV cccDNA inhibitor. J Hepatol 2023; 78: 742–53; Zoulim et al. Eliminating cccDNA to cure hepatitis B virus infection. J Hepatol 2023; 78: 677-80.
Response. Thank you, we included a new section 3.3.5 discussing small molecules including ccc_R08.
- Hepatitis delta affects roughly 5% of chronic hepatitis B patients globally. A short section discussing the challenges of this subset of patients would be worth to be included. The good news is that the achievement of functional HBV cure or, even better, eliminating/silencing HBV cccDNA could provide HDV cure indirectly. Check and refers to: Asselah et al. Hepatitis D virus infection.N Engl J Med. 2023; 389: 58-70.
Response. As suggested by the reviewer, we added the information on HDV to Introduction.
- Although the authors state that HBV prevention by expanding HBV vaccines would be the more efficacious way to reduce global HBV prevalence, two major challenges should be acknowledged. Firstly, HBV vaccine protection waning has been documented and may preclude long-lasting protection. In this regard, unexpected high rates of lack of HBV seroprotection have been found in recent surveys (i.e., Soriano et al. Susceptibility to hepatitis B virus infection in adults living in Spain.Liver Int. 2023; 43: 1015-20). Secondly, migration flows to Western countries (North America and EU) from HBV endemic regions in Asia and Africa contribute to keep stable HBsAg rates precluding the steadily declines expected because of universal HBV birth vaccination.
Response. Thank you, we included this to the Introduction and cited the provided references.
- Long-acting formulations of anti-HBV agents may be considered for indications other than treating chronic hepatitis B. This is the case for preventing HBV reactivations in patients undergoing immunosuppressive therapies or even hepatitis C treatment since HBV rebounds have been reported in these settings (especially in HBsAg+ but also in HBsAg-neg/anti-HBc+ individuals). Consider adding a paragraph discussing this topic at the end of section 4.2. Check and refers to: Soriano et al. Ultra-long-acting antivirals as chemical vaccines to prevent viral diseases.Future Microbiol. 2022; 17: 887-97.
Response. Thank you, we included this suggestion in section 4.1.
- English grammar is adequate and only needs minor revision and editing.
Response. The current version of the manuscript has been edited by English language professional, Grace Bybee, and we acknowledged her contribution.
Reviewer 3 Report
This review discusses new treatments for chronic hepatitis B virus infection with a focus on potential life-long therapies. The manuscript was well-written and definitely will contribute to the research field. They highlighted different classes of therapies, including entry inhibitors, capsid assembly modulators, and immunomodulators. More importantly, they also discussed the implants and gene editing approaches for the integrated viral genes and HBV cccDNA. I like this review a lot and do not have comments for the writing. Maybe the authors could somehow improve Figure 2 to make it clearer and better fit their highlights.
Author Response
I like this review a lot and do not have comments for the writing. Maybe the authors could somehow improve Figure 2 to make it clearer and better fit their highlights.
Response. We omitted multiple inhibitors of HBV replication in this figure to mainly focus prevention of the infectious particle’s generation and permanent destruction/silencing of cccDNA and integrated DNA as an important tool for HBV cure. However, based on the reviewer’s suggestion, we expanded this figure to show more treatment approaches.